

# Investigating molecular markers linked to acute myocardial infarction and cuproptosis: bioinformatics analysis and validation in the AMI mice model

Bingyu Wang, Jianqing Zhou and Ning An

Ningbo Medical Center Lihuili Hospital, Medical School of Ningbo University, Ningbo, Zhejiang, China

## ABSTRACT

Cuproptosis-related key genes play a significant role in the pathological processes of acute myocardial infarction (AMI). However, a complete understanding of the molecular mechanisms behind this participation remains elusive. This study was designed to identify genes and immune cells critical to AMI pathogenesis. Based on the GSE48060 dataset (31 AMI patients and 21 healthy persons, GPL570-55999), we identified genes associated with dysregulated cuproptosis and the activation of immune responses between normal subjects and patients with a first myocardial attack. Two molecular clusters associated with cuproptosis were defined in patients with AMI. Immune infiltration analysis showed that there was significant immunity heterogeneity among different clusters. Multiple immune responses were closely associated with Cluster2-specific differentially expressed genes (DEGs). The generalized linear model machine model presented the best discriminative performance with relatively lower residual and root mean square error, and a higher area under the curve (AUC = 0.870). A final two-gene-based generalized linear model was constructed, exhibiting satisfactory performance in two external validation datasets (AUC = 0.719, GSE66360 and AUC = 0.856, GSE123342). Column graph, calibration curve, and decision curve analyses also proved the accuracy of AMI prediction. We also constructed a mouse C57BL/6 model of AMI (3 h, 48 h, and 1 week) and used qRT-PCR and immunofluorescence to detect the expression changes of CBLB and ZNF302. In this study, we present a systematic analysis of the complex relationship between cuproptosis and a first AMI attack, and provide new insights into the diagnosis and treatment of AMI.

# INTRODUCTION

Increasing evidence indicates that inflammation and immunity play a critical role in atherosclerosis onset and progression (*Tan et al., 2011*; *Tang et al., 2015*; *Vellasamy et al., 2022*). Clinical manifestations are mainly caused by the formation of occluded thrombosis on the surface of ruptured or fissured plaque, rather than the narrowing of progressive lumen caused by slow-growing stable atherosclerotic plaques (*Formanowicz et al., 2019*).

Corresponding authors
Jianqing Zhou,
lhlzhoujianqing@nbu.edu.cn
Ning An, lhlanning@nbu.edu.cn

Despite increased identification of several circulatory markers associated with cardiovascular disease, biomarkers for the noninvasive diagnosis of atherosclerotic lesion rupture and markers of prognostic value for identifying individuals at high risk for future cardiovascular events are not yet available (*Wu et al., 2020*). Although significant contributions to subsequent plaque rupture have been ascribed to immune mechanisms and inflammatory status, few data sets have analyzed the correlation between acute myocardial infarction (AMI) and immune-related signaling molecular targets (*Gaisl et al., 2015*; *Ghasemzedah et al., 2017*). Furthermore, the validity of univariate biomarker prediction models have been questioned. Therefore, establishing multivariate prediction models for AMI at the molecular level and identifying molecular subtypes are of great clinical significance.

In the recent years, cuproptosis, a novel form of regulated cell death which is copper dependent has been identified, may be implicated in the process of various cardiovascular diseases (*Chen et al., 2023*; *Tsvetkov et al., 2022*). The micronutrient copper plays an important role in maintaining human homeostasis and has antioxidant properties (*Uriu-Adams & Keen, 2005*). Both superoxide dismutase 1 (SOD1) and cytochrome C oxidase (CytoC) in mitochondria require the involvement of copper (*Flores-Cotera et al., 2021*). Copper is closely related to the pathophysiological processes of the cardiovascular system, so it is necessary to further elucidate the molecular characteristics of cuproptosis-related genes (CRGs) that may explain the cause of AMI heterogeneity (*Fukai, Ushio-Fukai & Kaplan, 2018*).

This study marks the first time that CRG and immune signature expression in normal and AMI individuals were systematically compared. We divided 31 AMI patients into two clusters associated with cuproptosis, based on the expression landscapes of seven CRGs, and then analyzed the immune cell differences between the two groups. The weighted gene co-expression network analysis (WGCNA) algorithm was used to identify cluster-specific differentially expressed genes (DEGs), and its rich biological functions and pathways were then elucidated based on these DEGs. Additionally, a predictive model was developed by comparing multiple machine learning algorithms to identify patients with different molecular clusters. Column diagrams, calibration curves, and decision curve analysis (DCA) were used to predict the performance of the model. Finally, we investigated the sensitivity of different genes related to cuproptosis in non-AMI patients as well as in AMI patients, thus providing new insights into the diagnosis and treatment of AMI.

## MATERIALS

### Data acquisition

A microarray dataset (GSE48060) related to AMI was obtained from the GEO website database (GEO, https://www.ncbi.nlm.nih.gov/geo/) using the "GEOquery" R package (version 4.1) (*Davis & Meltzer, 2007*). The GSE48060 dataset (GPL570-55999 platform) including 21 healthy and 31 AMI blood samples were selected for further analysis. Robust multiarray average (RMA) was used to normalize the raw gene expression profiles of the GEO dataset ("affy", version 4.1.0 of the R package).

## Animals and procedures

For animal models of MI, male C57BL/6 mice at the age of 6–7 weeks were purchased from Ningbo University. The study was approved by the use committee at Ningbo University School of Medicine (11649) and conducted in accordance with the National Institutes of Health Guide. Animals were kept at $23 \pm 2$ °C with $50 \pm 5\%$ humidity. Mice were randomly split into two groups ($n = 10$ each): sham and MI. Randomization was done using the Rand() function in Excel based on weight and cage position. Select one cage at random from all the cages in each group. Two animals are taken from each individual recombination and given a permanent number in the cage. The cages were then randomly assigned to the experimental group. The specific steps of MI mouse model establishment are in the Supplemental file S4. The experimental mice were euthanized according to the experimental endpoint set in the experiment. We sacrificed mice after 3 h, 48 h, and 1 week from the beginning of the animal model. Stack experimental animals in the euthanasia box, and introduce carbon dioxide to the animal. There were no mice left in this study and were used for experiments. After 3 h, 48 h, 1 week of MI, the mice were anaesthetized and sacrificed for collection of heart tissues and follow-up examinations.

## Echocardiographic analysis

Echocardiography was performed on mice 2 weeks after MI or sham surgery. The mice were anesthetized with isoflurane and their cardiac function was evaluated using a small animal ultrasound system (Vevo 2100; VisualSonics, Toronto, Canada). Cardiac function indicators, such as left ventricular ejection fraction and fractional shortening, were calculated and averaged from at least three cardiac cycles by a single technician in a blinded manner.

## Immunofluorescence

Heart sections were deparaffinized, rehydrated, and subjected to antigen retrieval using 10 mM citrate buffer (pH 6.0) at 95 °C for 10 min. After washing with PBS, the sections were permeabilized with 0.5% Triton X-100, blocked with 5% BSA, and incubated with primary antibodies overnight at 4 °C, followed by incubation of secondary antibodies. DAPI was used to label cell nuclei before imaging with a Leica SP8 confocal microscopy.

The primary antibodies used were as follows: ZNF302 Polyclonal Human Antibody, 1:100, (Thermo Fisher Scientific, Waltham, MA, USA); CBLB Monoclonal Mouse Antibody, 1:100, (Abcam, Cambridge, UK); Abclonal, FITC Goat Anti-Rabbit IgG (H+L) (AS011) 1:100; and FITC Goat Anti-Mouse IgG (H+L) (AS001), 1:100.

## Quantitative reverse transcription-polymerase chain reaction

The mouse was sacrificed and serum was collected. Total RNA was extracted using the Trizol method. Reverse transcription was performed with a kit from TransGen Biotech (Beijing, China). The cDNA was amplified using real-time PCR and normalized to the housekeeping gene GAPDH. The CT values were analyzed using Applied Biosystems Software, and relative mRNA expression was calculated using $2^{-\Delta\Delta C}$. The sequences of the primers are following: CBLB (5′-3′): Forward GGTCGCATTTTGGGGATTATTGA,
Reverse TTTGGCACAGTCTTACCACTTT; ZNF302 (5′-3′): Forward ATCCTCTATT CCTGTTCCTGGGA, Resverse CAGGCCAAATAACATAGGGAGAC.

### 2,3,5-triphenyltetrazolium chloride stain

Following a myocardial infarction event, the animals underwent anesthesia, intubation, and thoracotomy. The hearts were excised, and the left ventricles were then precisely sliced into 1 mm thick sections. Heart sections were then incubated with a TTC solution (T130066-5 g; Aladdin, Shanghai, China).

### Assessing immune cell infiltration

We used a combination of the CIBERSORT algorithm (https://cibersort.stanford.edu/) and Leukocyte signature matrix22 (LM22) to generate this signature matrix and estimate the relative abundance of 22 types of immune cells in each sample. CIBERSORT calculates inverse fold product $p$-values using Monte Carlo sampling. Samples with $p$-values less than 0.05 were considered accurate immune cell fractions. On the basis of the sum of the 22 proportions, each sample contained one immune cell (*Newman et al., 2015*). Reference data sets we put in supplementary data.

### Immune cell infiltration and CRG correlation analysis

In order to further demonstrate the association between CRGs and AMI-associated immune cell properties, CRG expression was correlated with immune cell percentage. Statistical significance was determined using the Spearman correlation coefficient when the $p$-value was less than 0.05. Corrplot (version 0.92) was then used to display the results.

### Clustering of AMI patients without supervision

Initially, a total of seven CRGs were obtained according to the previous study by *Tsvetkov et al. (2022)*. The 31 AMI samples were classified into different clusters by unsupervised clustering (Consensus Cluster Plus, version 4.2.1) using seven CRG expression profiles and the k-means algorithm with 1,000 iterations (*Langfelder & Horvath, 2008*). The maximal subtype number k (k = 2) was chosen, and the cluster number was determined using the cumulative distribution function (CDF) curve, consensus matrix, consistent cluster number, and consistent cluster score ($p > 0.9$).

### Gene set variation analysis

Gene set variation analysis (GSVA) was performed using the R package "GSVA" (version 4.2.1) to examine the differences in enriched gene sets between different clusters of CRGs. The "c2.cp.kegg.symbols.gmt" and "c5.go.symbols.gmt" files were obtained from the MSigDB website database for further GSVA. GSVA scores were compared between different CRG clusters using the "limma" R package (version 4.2.1) and were considered significant if the |t value of GSVA score| was more than 2.

### Weighted gene co-expression network analysis

Weighted gene co-expression network analysis (WGCNA) was performed to identify co-expression modules using the R package of "WGCNA" (version 4.2.1). To ensure

accuracy, we used the top 25% of genes with the highest variance for subsequent WGCNA analyses. To construct a topological overlap matrix (TOM), we constructed a weighted adjacency matrix with an optimal soft power. As the minimum module size was set to 100, dissimilarity measures (1-TOM) were computed based on the hierarchical clustering tree algorithm. Modules were assigned random colors. In each module, a module eigengene represented global gene expression profiles. Modular significance (MS) demonstrated the relationship between modules and disease. Clinical phenotype and gene significance (GS) have been described as correlated processes in biology.

## Machine learning-based prediction model construction

The "caret" R packages (version 6.0.93) were used to establish machine learning models (random forest model (RF), support vector machine model (SVM), generalized linear model (GLM), and Xtreme Gradient Boosting (XGB)) based on two different CRG clusters. As an ensemble machine learning approach, RF utilizes multiple independent decision trees to predict classifications or regressions (*Rigatti, 2017*). The SVM algorithm generates a hyperplane in the characteristic space that distinguishes between positive and negative instances with a maximum margin (*Tang, Tian & Yang, 2018*). A GLM is a multivariate linear regression method that can be used to analyze relationships between categorical or continuous factors and normally distributed dependent factors (*Gold & Sollich, 2003*). Based on gradient boosting, XGB compares classification error and complexity by comparing boosted trees (*Sheridan et al., 2020*). DEGs were used as explanatory variables in this study in conjunction with distinct clusters as response variables. The 31 AMI samples were randomly classified into a training set (70%, $N = 22$) and a validation set (30%, $N = 9$). Grid search was used by the caret package to tune parameters in these models, and all models were evaluated with default parameters using the 5-fold cross-validation method. Using the "pROC" R package (version 1.18.0), we visualized the area under ROC curves. Consequently, the top five predictive genes were determined to be the most important variables for predicting AMI. For verification of the model's diagnostic value, ROC curves were examined in datasets (GSE123342 and GSE66360).

## Model construction and validation for nomograms

The "rms" R package (version 6.2.0) was used to develop a nomogram model to identify AMI clusters. Scores are assigned to each predictor, and the "total score" represents the sum of the scores. Decision curve analysis (DCA) is a method for evaluating clinical predictive models, diagnostic tests, and molecular markers (*Ji & Xue, 2020*). Nomogram models were evaluated using the calibration curve and DCA.

## Drug-therapy analysis

Drug data were downloaded from the Drug Gene Interaction Database (DGIdb) website (https://dgidb.org/search_interactions). We used two hub genes to predict the underlying drug.

## Statistical analysis

All data are presented as the mean ± standard deviation. GraphPad Prism 8.0 software (Version X; La Jolla, CA, USA) was used to analyse the data. Unpaired two tailed Student's t tests and one-way ANOVA were used for data analysis followed by *post hoc* tests. $p < 0.05$ was considered statistically significant.

## RESULTS

### Activation of immune responses in AMI patients due to dysregulation of cuproptosis regulators

To elucidate the biological functions of regulatory factors in the occurrence and development of AMI, the GSE48060 dataset was used to systematically evaluate the differences in expression profiles of 46 CRGs between AMI and non-AMI patients. Figure 1 shows a detailed flow chart of the research process. A total of seven CRGS were identified as differentially expressed cuproptosis genes. Among them, the expression levels of metal regulatory transcription factor 1 (MTF1) and antioxidant 1 copper chaperone (ATOX) were higher, while the expression levels of mitogen-activated protein kinase kinase 1 (MAP2K1), Phosphodiesterase 3B (PDE3B), lipoyltransferase 1 (LIPT1), glutaminase (GLS), and dihydrolipoamide branched chain transacylase E2 (DBT) genes in AMI serum were much lower than those in the non-AMI control group base on log2 fold change (Figs. 2A–2C). Our next step was to examine whether cuproptosis regulatory factors played a fundamental role in the progression of AMI. Surprisingly, some cuproptosis regulators, such as DBT and GLS, showed strong synergies (coefficient = 0.84). At the same time, LIPT1 and MTF1 showed obvious antagonism (coefficient = 0.59). Furthermore, ATOX1 and DBT were associated with other regulators (Fig. 2D) based on the correlation profiles of these CRGs. To clarify whether there were differences in the immune system between AMI and normal controls, immune-infiltration analysis was performed to determine the proportion of 22 immune cell types infiltrated between AMI and non-AMI control subjects according to the CIBERSORT algorithm (Fig. 2E). The results showed that there were significant differences in T cells CD4 memory activated, NK cells resting, macrophages M0, and neutrophils in patients with AMI (Fig. 2F), indicating that the alternation of the immune system may be the main cause of AMI. At the same time, correlation analysis showed that B cells memory, dendritic cells resting, macrophages M0, mast cells activated, monocytes, neutrophils, NK cells resting, T cells CD4 memory resting, T cells CD4 native, and T cells gamma delts were all associated with cuproptosis regulatory genes (Fig. 2G). These results suggested that CRGs may be the critical factors involved in regulating the molecular and immune infiltration status of AMI patients.

### Identifying cuproptosis clusters in AMI

To explore the expression patterns associated with cuproptosis in AMI, we used a consensus clustering algorithm to group 31 AMI samples according to the expression profiles of 46 CRGs. A total of seven CRGs were determined as differentially expressed cuproptosis genes. When the k value was set to 2 (k = 2), the cluster number was the most stable, and the consensus index of CDF curve ranged from 0.2 to 0.6 (Figs. 3A, 3B). When

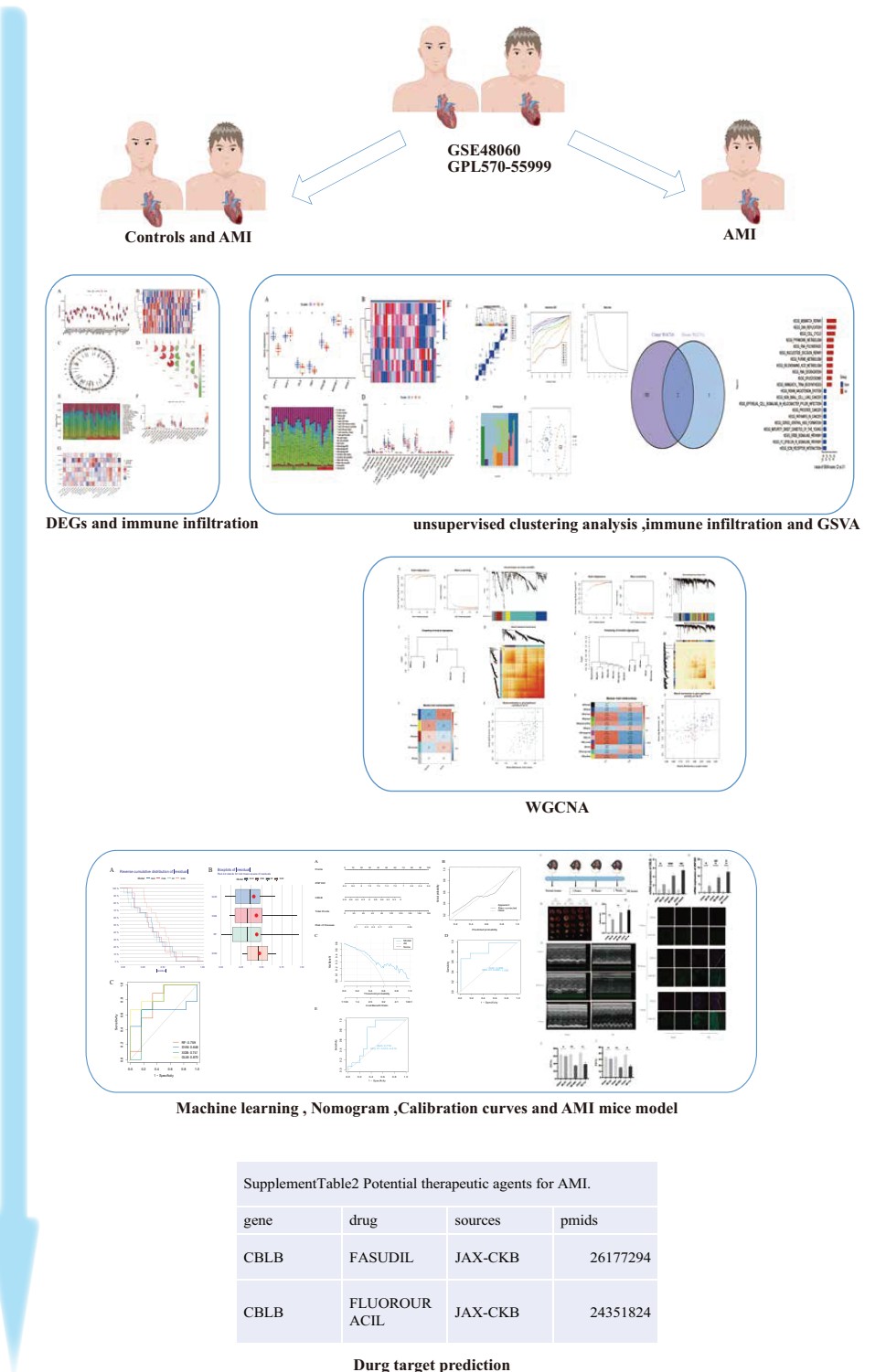

Controls and AMI

GSE48060
GPL570-55999

AMI

**DEGs and immune infiltration**

**unsupervised clustering analysis ,immune infiltration and GSVA**

**WGCNA**

**Machine learning , Nomogram ,Calibration curves and AMI mice model**

| SupplementTable2 Potential therapeutic agents for AMI. | | | |
|---|---|---|---|
| gene | drug | sources | pmids |
| CBLB | FASUDIL | JAX-CKB | 26177294 |
| CBLB | FLUOROUR ACIL | JAX-CKB | 24351824 |

**Durg target prediction**

**Figure 1  The study flow chart.** The human and heart graphs were provided by Figdraw (https://www.figdraw.com/#/; Authorization ID: SORP49ad0).  

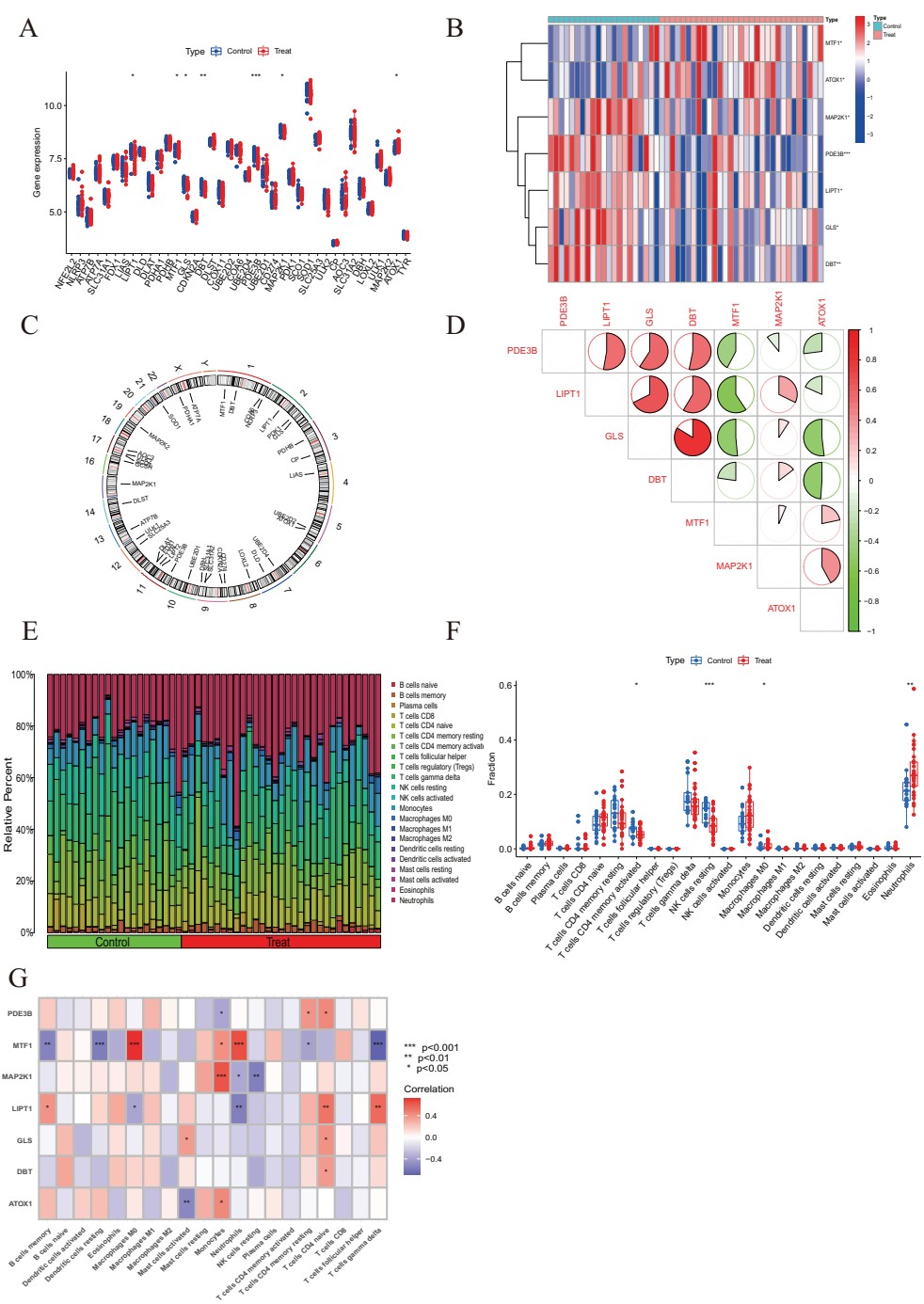

**Figure 2 Identification of dysregulated CRGs in AMI.** (A) The expression patterns of seven CRGs were presented in the heatmap. (B) Boxplots showed the expression of seven CRGs between AMI and non-AMI controls. (C) The location of 45 CRGs on chromosomes. (D) Correlation analysis of seven differentially expressed CRGs. Green and red colors represent negative and positive correlations, respectively. The correlation coefficients were marked with the area of the pie chart. (E) The relative abundances of 22 infiltrated immune cells between AMI and non-AMI controls. (F) Boxplots showed the differences in immune infiltrating between AMI and non-AMI controls. (G) Correlation analysis between seven differentially expressed CRGs and infiltrated immune cells. $^*p < 0.05$, $^{**}p < 0.01$, $^{***}p < 0.001$.

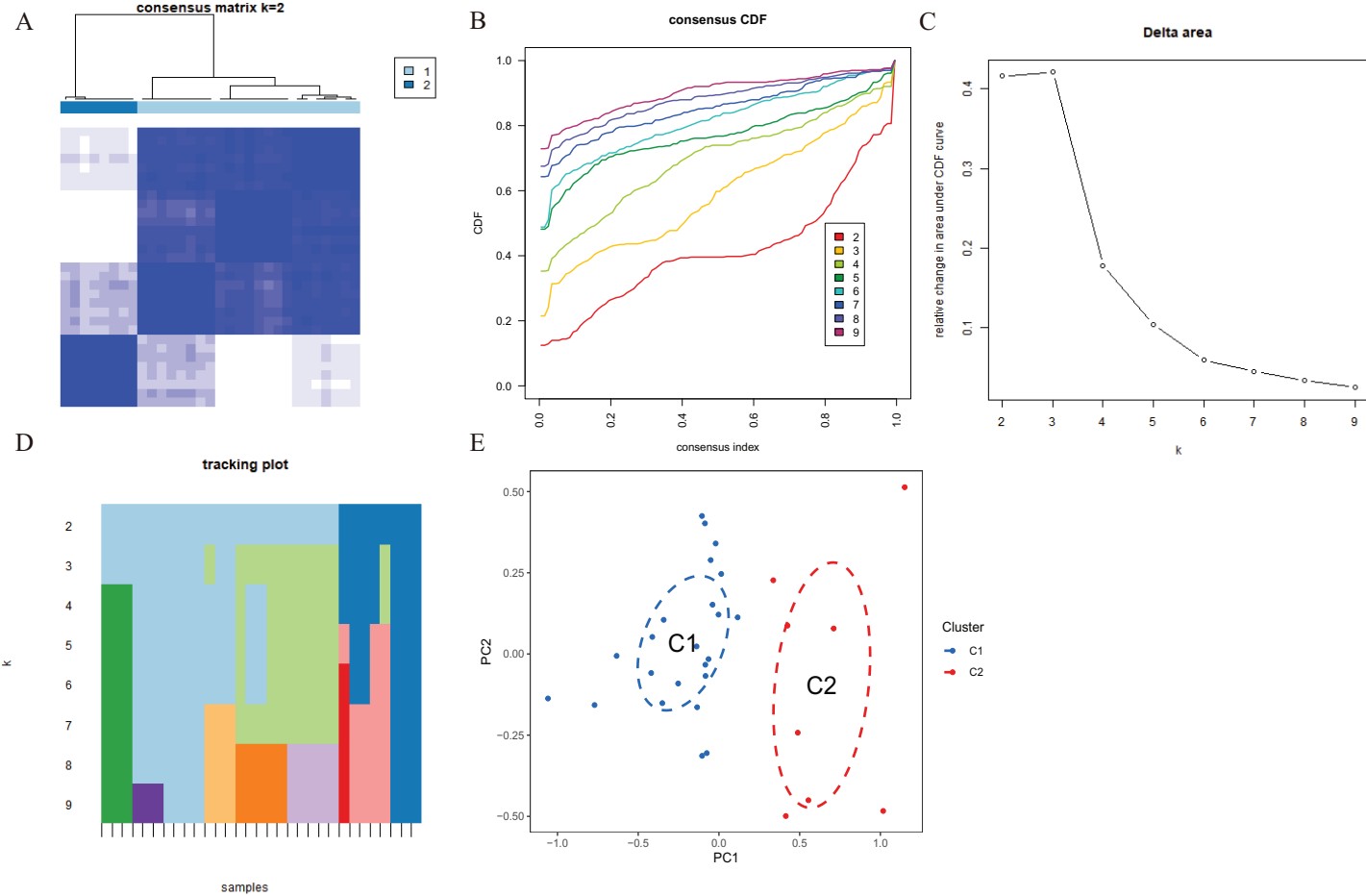

**Figure 3  Identification of cuproptosis-related molecular clusters in AMI.** (A) Consensus clustering matrix when k = 2. (B–D) Representative cumulative distribution function (CDF) curves (B), CDF delta area curves (C), the score of consensus clustering (D). (E) t-SNE visualizes the distribution of two subtypes.

k = 2 to 6 (k = 2–6), a difference between the two CDF curves could be seen in the area under the CDF curve (k and k-1) (Fig. 3C). Additionally, only k = 2 resulted in a consistency score of >0.9 for each subtype (Fig. 3D). Combined with the consensus matrix table, 31 AD patients were finally divided into two clusters, Cluster 1 (*n* = 23) and Cluster 2 (*n* = 8) (Table S1). According to tSNE analysis (Fig. 3E), there were significant differences between the two clusters.

## Characteristics of cuproptosis clusters and cuproptosis regulators

A comprehensive evaluation of the expression differences between Cluster 1 and Cluster 2 was conducted first in order to investigate the molecular characteristics between clusters. Different CRG expression levels were assessed between the two cluster modes. Cuproptosis Cluster 1 showed high expression levels of PDE3B, LIPT1, GLS, and DBT, while cuproptosis Cluster 2 was characterized by high expression levels of MTF1, MAP2K1, and ATOX1 (Figs. 4A, 4B). In addition, immune infiltration analysis showed that Cluster 1 and Cluster 2 had a high proportion of CD8+T cells, T cells CD4 live, T cells CD4 memory resting, T cells CD4 memory activated in cuproptosis (Figs. 4C, 4D). The abundance of monocells and Neutrophils in Cluster 2 was relatively high. This suggests that cuproptosis

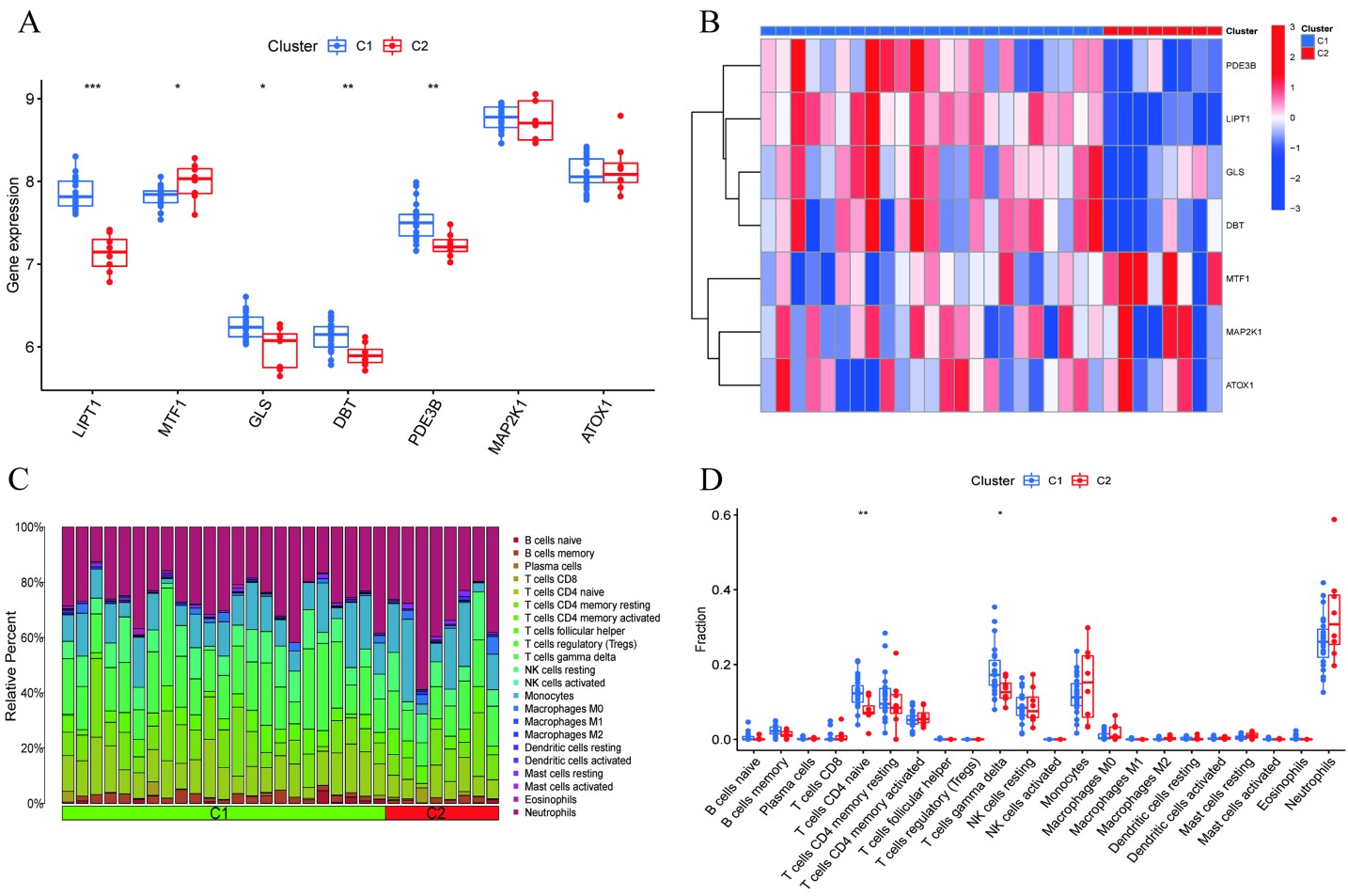

**Figure 4** **Identification of molecular and immune characteristics between the two cuproptosis clusters.** (A) Boxplots showed the expression of seven CRGs between two cuproptosis clusters. (B) Clinical features and expression patterns of 7 CRGs between two cuproptosis clusters were presented in the heatmap. (C) Boxplots showed the differences in immune infiltrating between two cuproptosis clusters. (D) The relative abundances of 22 infiltrated immune cells between two cuproptosis clusters. $^*p < 0.05$, $^{**}p < 0.01$ $^{***}p < 0.001$.

Cluster 1 may have a more dominant level of immune infiltration. Using WGCNA, we established co-expression networks and modules for normal and AMI subjects to identify hub genes. GSE48060 variance was calculated for each gene expression, and the top 25% of genes with the highest variance were further analyzed. Soft power value 12 and no scale R led to the identification of co-expressed gene module 2 equal to 0.9 (Fig. 5A). The dynamic cutting algorithm was used to obtain five different co-expression modules with different colors, and the heat map of the TOM was constructed (Figs. 5B–5D). The modular-clinical feature (control and AMI) co-expression was subsequently analyzed using these genes in the five color modules. AMI was most strongly related to the blue module, which included 796 genes (Fig. 5E). Further, yellow module genes correlated positively with mode-related genes (Fig. 5F). In addition, we used the WGCNA algorithm to analyze the key gene modules closely related to cuproptosis clusters. We screened β = 17 and R² = 0.9 to construct the scale-free network (Fig. 6A). In particular, 12 modules containing 5,413 genes were identified as important modules, and the heat map illustrated the TOMs of all

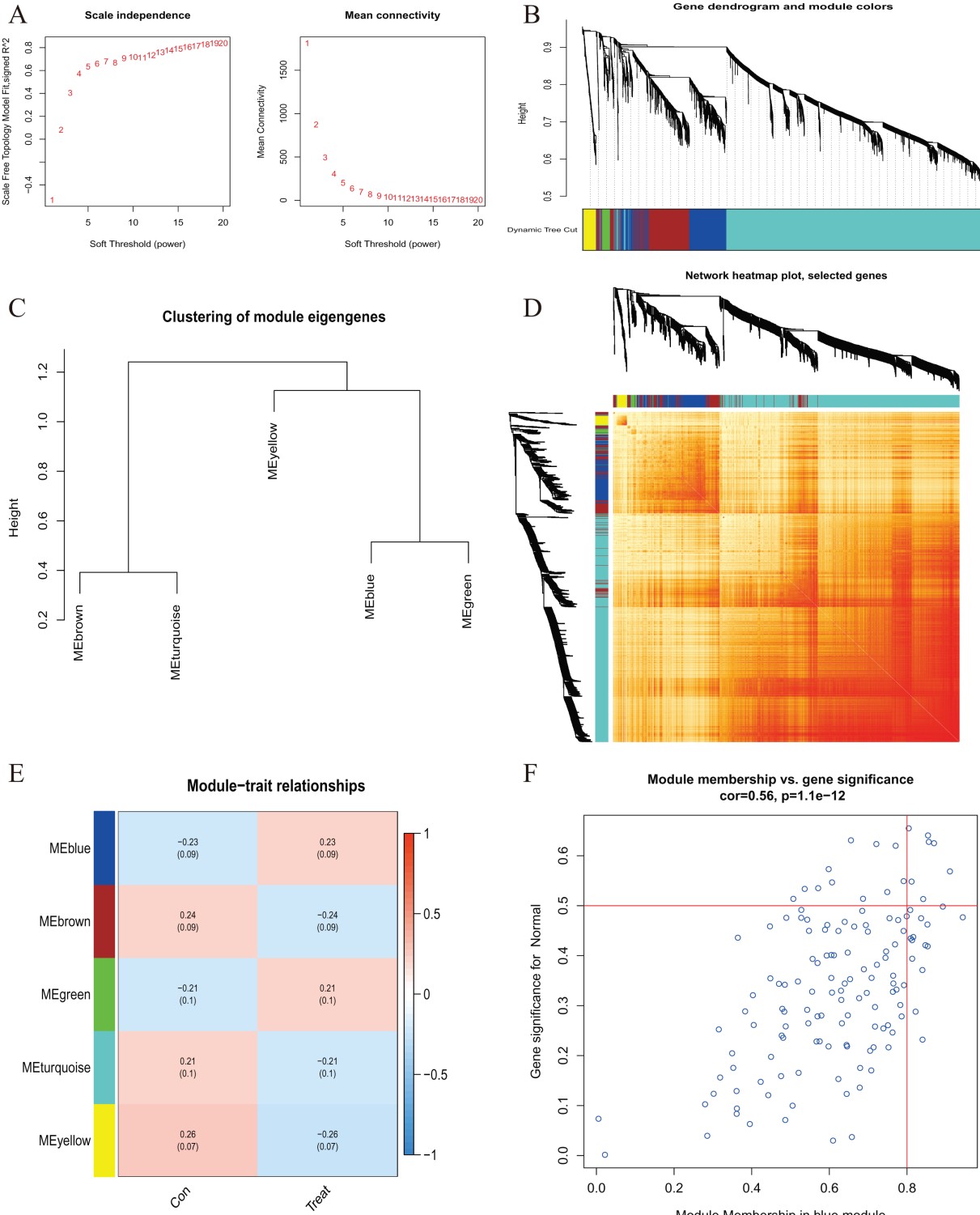

**Figure 5** **Co-expression network of differentially expressed genes in AMI.** (A) The selection of soft threshold power. (B) Cluster tree dendrogram of co-expression modules. Different colors represent distinct co-expression modules. (C) Representative of clustering of module eigengenes. (D) Representative heatmap of the correlations among five modules. (E) Correlation analysis between module eigengenes and clinical status. Each row represents a module; each column represents a clinical status. (F) Scatter plot between module membership in blue module and the gene significance for AMI.

genes related to these modules (Figs. 6B–6D). Analysis of modular-clinical features (Clusters 1 and 2) showed that the amethyst module (1,115 genes) was associated with AMI clusters (Fig. 6E). As a result of correlation analysis, the gene for the amethyst module was associated with the selected module (Fig. 6F).

## Cluster-specific DEG identification and functional annotation

A total of 188 cluster-specific DEGs were identified by analyzing the intersection of cuproptosis module related genes with AMI and non-AMI individual module-related genes (Fig. 7A). GSVA was used to further explore functional differences between the two clusters related to cluster-specific DEGs. The results showed that Mismatch repair, DNA replication, cell cycle, pyrimidine metabolism, RNA polymerase, nucleotide excision repair, and purine metabolism were all present in Cluster 2. However, ECM preceptor interaction and FC epsilon RI signaling pathway were up-regulated in Cluster 1 (Fig. 7B). Therefore, a number of immune responses and metabolic processes may be mediated in Cluster 2.

## Machine learning model construction and assessment

We developed four validated machine learning models, RF, SVM, GLM, and XGB, based on the expression profiles of 52 cluster-specific DEGs in the AMI cohort. To interpret the four models and plot the residual distribution for each model in the test set, we applied the "DALEX" package. The GLM machine learning model had relatively low residuals (Figs. 8A, 8B). On the basis of five-fold cross validation, we calculated the ROC curve for four machine learning algorithms. The ROC curve showed the maximum area under general linear model (GLM, AUC = 0.870; RF, AUC = 0.759; SVM, AUC = 0.648; XGB, AUC = 0.741, Fig. 8C). Based on these results, the GLM showed the best ability to discriminate between patient clusters. Finally, the two most important variables in the GLM (ZNF302, CBLB) were selected as predictive genes for further analysis. The GLM model was then evaluated using a column graph to estimate 31 AMI patients (Fig. 9A). Line graph models were evaluated using calibration curves and DCA. AMI cluster risks and predicted risks are very close based on the calibration curve (Fig. 9B). The DCA indicates the high accuracy of our column graph, which may provide a basis for clinical decision-making (Fig. 9C). ROC curves showed satisfactory performance of the 2-gene prediction model with an AUC value of 0.856 in the GSE123342 (Fig. 9D) and 0.719 in the GSE66360 dataset (Fig. 9E), indicating our diagnosis model is equally efficacious in distinguishing AMI from normal individuals. DGIdb was used for drug prediction of the hub genes (Table S2). There was a positive finding for Fasudil and Fluorouracil, but both of these drugs are new and need further research before they can be used clinically.

## Expression of the two genes in AMI tissues

To further verify the expression of these two genes *in vivo*, 6-week-old C57BL/6 mice were divided into sham and AMI groups. Considering the crucial time intervals associated with myocardial infarction, we established three specific time points for our study. The onset of acute myocardial infarction occurs at the 3-h mark, while the 48-h mark represents the

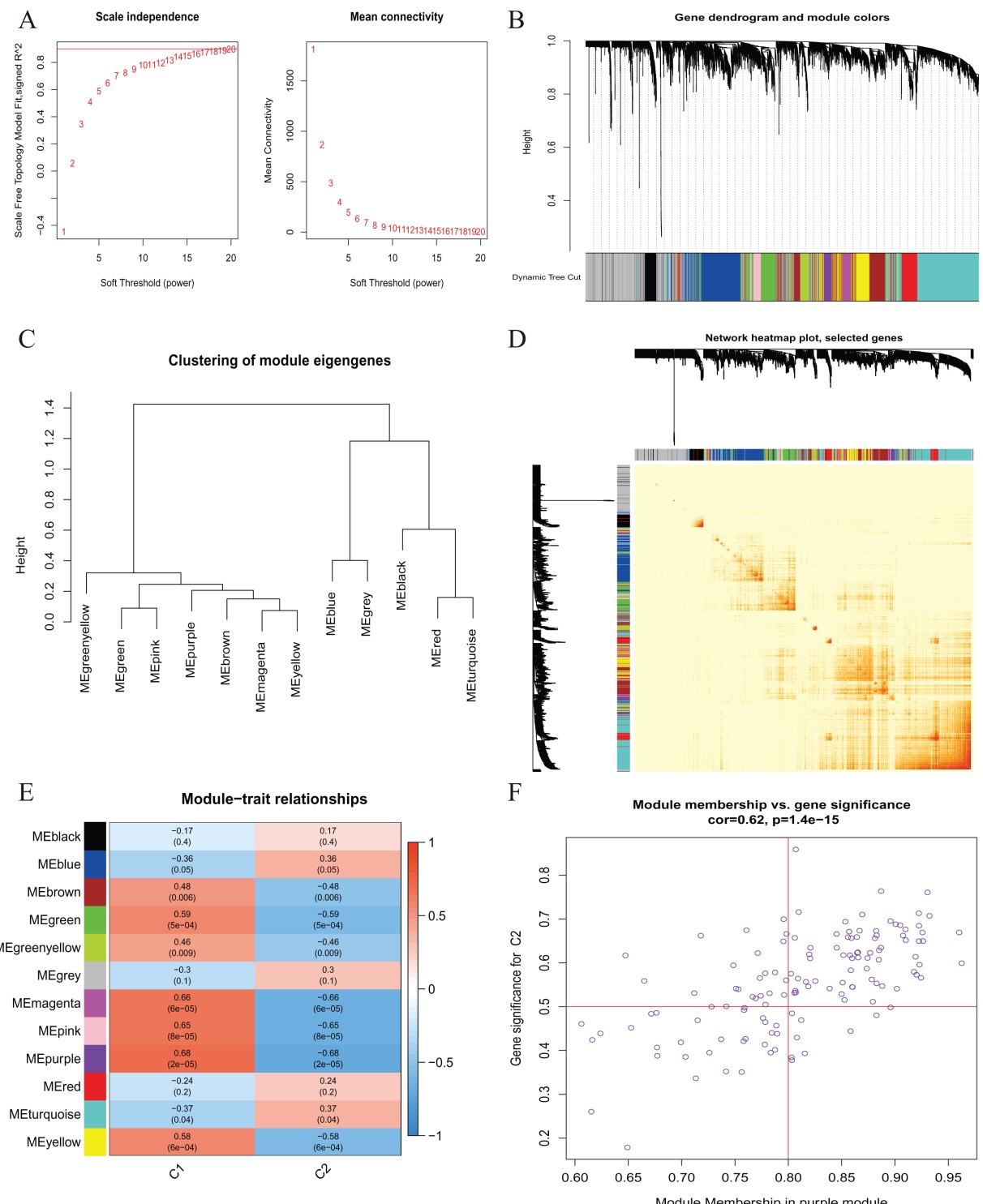

**Figure 6 Co-expression network of differentially expressed genes (DEGs) between the two cuproptosis clusters.** (A) The selection of soft threshold power. (B) Cluster tree dendrogram of co-expression modules. Different colors represent distinct co-expression modules. (C) Representative of clustering of module eigengenes. (D) Representative heatmap of the correlations among 12 modules. (E) Correlation analysis between module eigengenes and clinical status. Each row represents a module; each column represents a clinical status. (F) Scatter plot between module membership in purple module and the gene significance for Cluster 1.

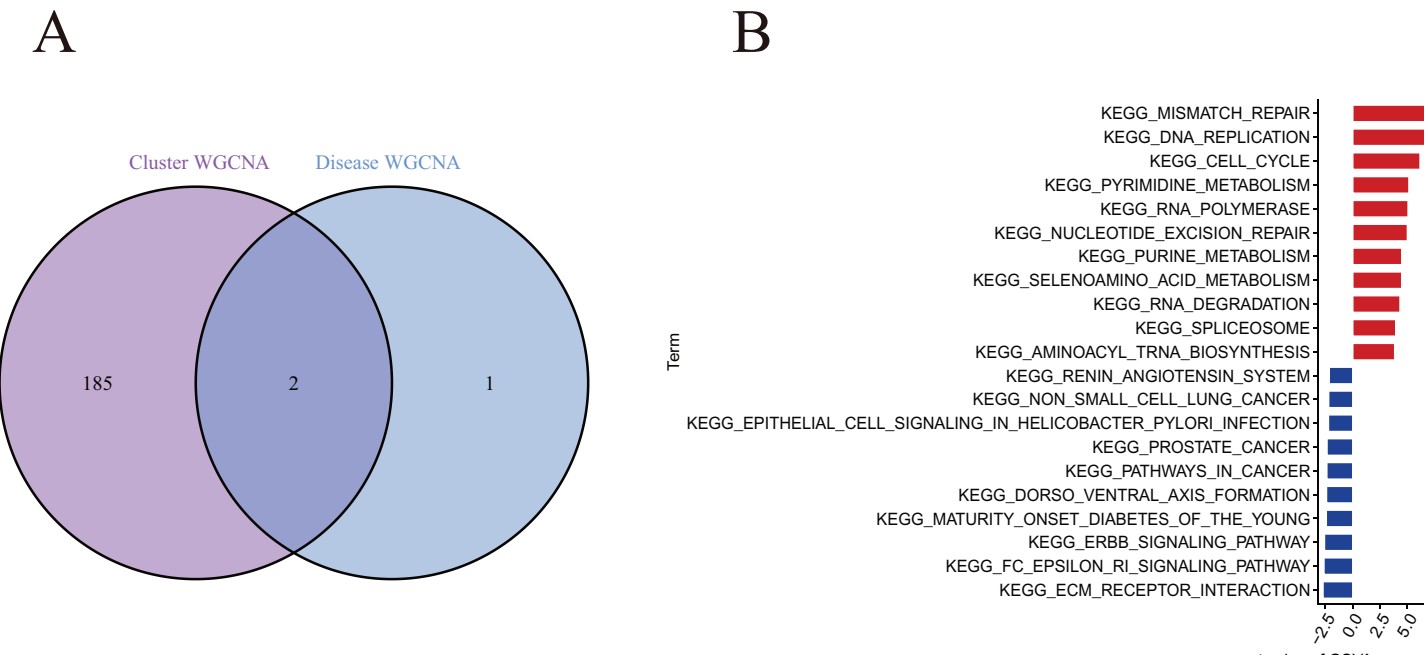

**Figure 7  Identification of cluster-specific DEGs and biological characteristics between two cuproptosis clusters.** (A) The intersections between module-related genes of cuproptosis clusters and module-related genes in the GSE48060 dataset. (B) Differences in hallmark pathway activities between Cluster 1 and Cluster 2 samples ranked by t-value of GSVA method.               

timeframe beyond which surgical intervention for acute myocardial infarction is no longer optimal. During this period, we examined the expression of genes related to myocardial infarction. Additionally, we utilized mice with myocardial infarction for a duration of 1 week to investigate changes in molecular expression levels associated with cuproptosis as myocardial infarction progresses. A schematic diagram shows the animal experiment design (Fig. 10A). Additionally, the results of TTC staining were used to verify the degree of ischemic infarction in mice (Figs. 10B, 10C). Based on the statistical data pertaining to myocardial damage, TTC staining revealed a degree of myocardial damage after a duration of 3 h, albeit not as pronounced as the myocardial damage observed after 48 h and 1 week (Fig. 10C). Additionally, echocardiography was employed to identify alterations in cardiac function among mice. The findings indicated that the ejection and contractile functions of the heart exhibited no significant reduction 3 h post AMI. However, a notable decrease was observed at the 48 h and 1 week marks following MI, as illustrated in Figs. 10D–10F. The serum of mice was collected at three different time points for analysis, revealing an increase in the expression of CBLB and ZNF302 at 3 h post-myocardial infarction. The expressions of CBLB and ZNF302 in serum collected at 48 h and 1 week post-myocardial infarction were significantly higher compared to the sham operation group (Figs. 10G, 10H). The immunoconfocal technique was employed to examine the fluorescence staining outcomes of myocardial sections from mice. Our findings indicate that there was no notable disparity in the expression of the two proteins in mice 3 h following MI (Fig. 10I). Nevertheless, disparities in both markers were observed in the myocardial sections of mice 48 h after MI, revealing an increase in marker expression within the tissue subsequent to MI (Fig. 10I).
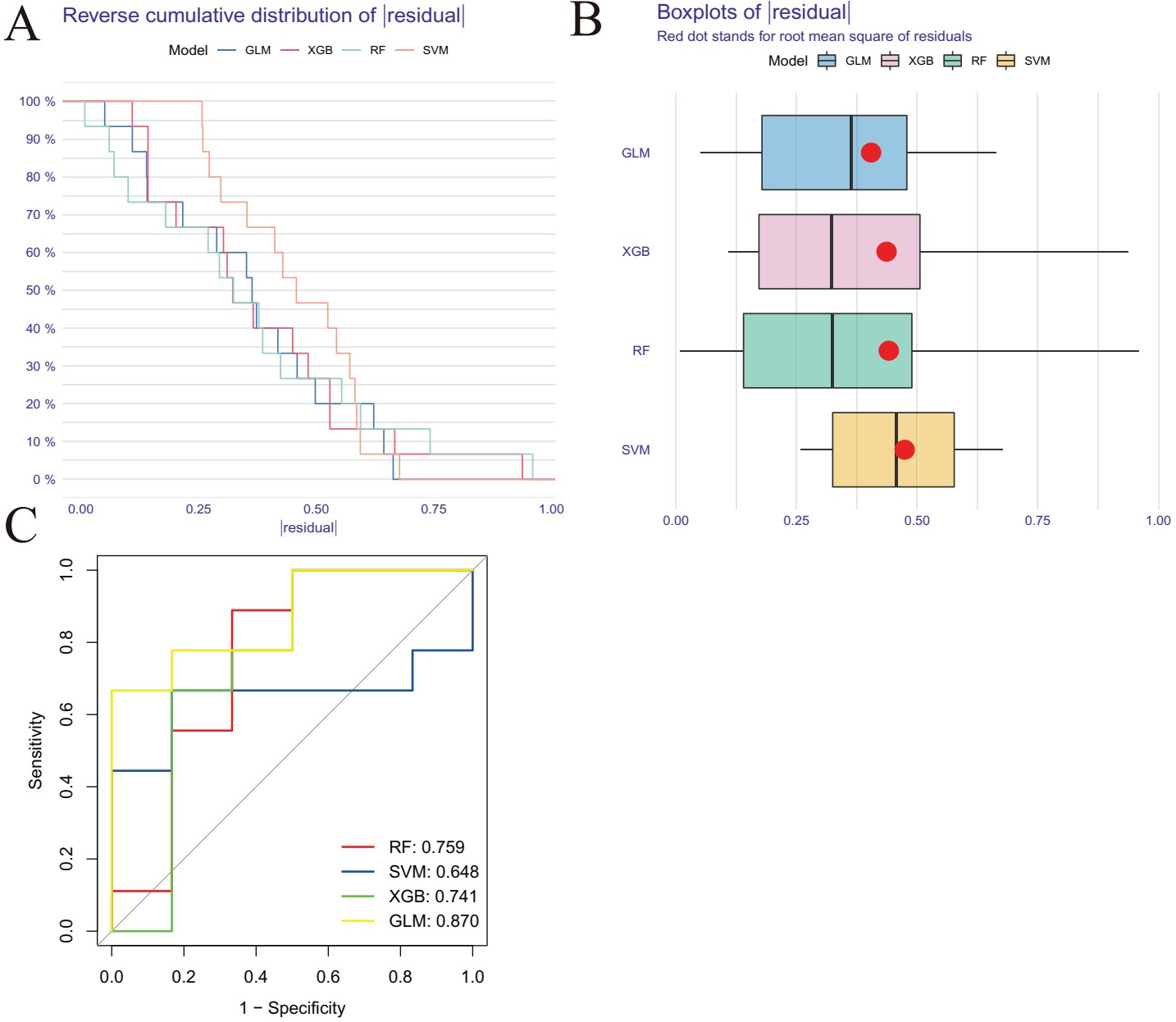

**Figure 8 Construction and evaluation of RF, SVM, GLM, and XGB machine models.** (A) Cumulative residual distribution of each machine learning model. (B) Boxplots showed the residuals of each machine learning model. Red dot represented the root mean square of residuals (RMSE). (C) ROC analysis of four machine learning models based on five-fold cross-validation in the testing cohort.

Notably, the disparity between these two expressions was more pronounced in mice with continuous MI for 1 week (Fig. 10I). These findings suggest that the expression difference of these two molecules is not prominent in the early stages of myocardial infarction, but becomes more pronounced over time. These two molecules may serve as potential diagnostic markers for myocardial infarction, particularly left ventricular infarction. Further research is needed to elucidate their role in the treatment of myocardial infarction

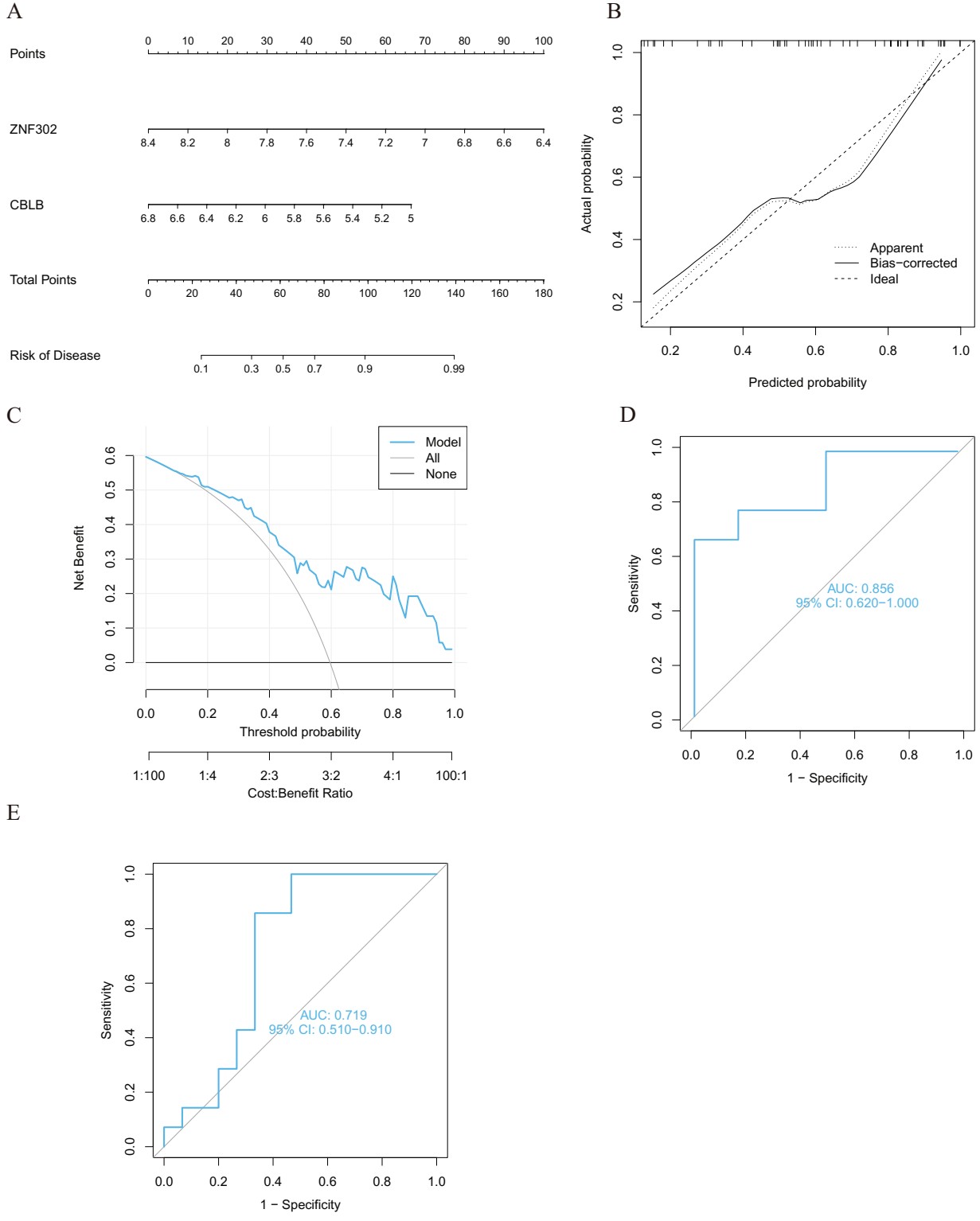

**Figure 9 Validation of the two-gene-based GLM model.** (A) Construction of a nomogram for predicting the risk of AMI clusters based on the two-gene-based GLM model. (B, C) Construction of calibration curve (B) and DCA (C) for assessing the predictive efficiency of the nomogram model. (D) ROC analysis of the two-gene-based GLM model based on five-fold cross-validation in GSE123342 dataset. (E) ROC analysis of the 2-gene-based GLM model based on five-fold cross-validation in GSE66360 dataset.

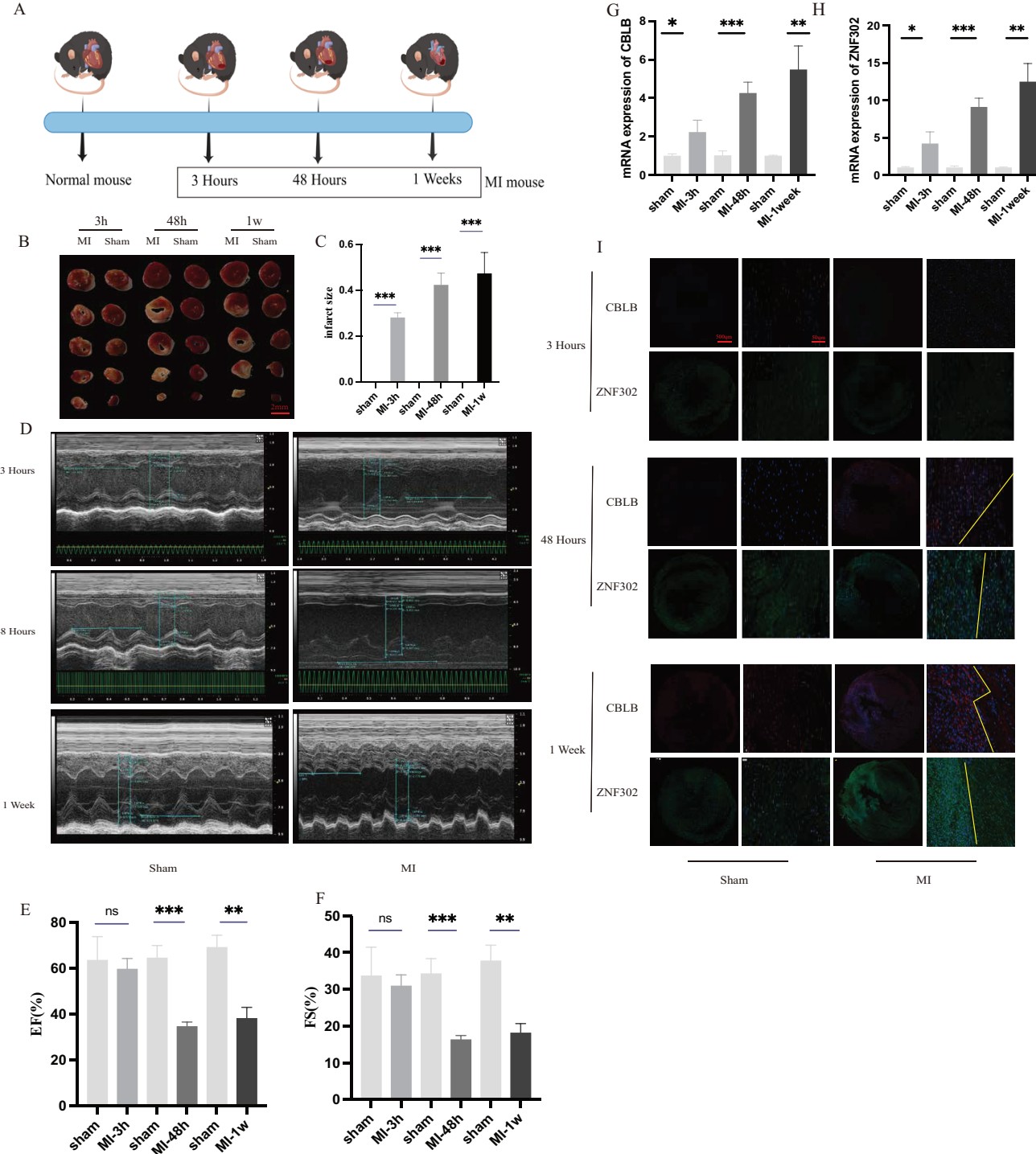

**Figure 10 The expression of ZNF302 and CBLB in sham and MI model mice.** (A) Schematic diagram demonstrates the animal experiment design. (B) TTC staining was performed to detect cardiac ischemic area of different group mice. Scale bar = 2 mm. $n$ = 4 per group. (C) Quantitative analysis of cardiac ischemic area. (D) Representative M-mode echocardiographic recording obtained from different groups mice. $n$ = 3 per group. (E and F) Quantitative analysis of LV ejection fraction (EF), fractional shortening (FS). (G and H) The mRNA expression levels of CBLB and ZNF302 were analysed by qRT-PCR, number = 3. (I) Representative immunofluorescence (green for ZNF302, red for CBLB, blue for DAPI). $n$ = 3 per group. The mice and heart graphs were provided by Figdraw (https://www.figdraw.com/#/; Authorization ID: SORUROde70). $^{**}p < 0.001$, $^{***}p < 0.001$, ns, no significance.

and the specific mechanism by which they exert their immune effects. These results suggest that ZNF302 and CBLB are risk factors for the development of AMI in mice, although its specific mechanism in AMI patients requires further study.

## DISCUSSION

In the current treatment of AMI, drug and reperfusion therapy are mainly used because of the heterogeneity of the pathology (*Collet et al., 2021*; *Ibanez et al., 2018*). Even though treatments for MI have advanced greatly in recent decades, new therapies for the condition (such as cell and gene therapy) are still under investigation (*Collet et al., 2021*; *Ibanez et al., 2018*). The urgent need for improved treatment necessitates the identification of more suitable molecular clusters, which is crucial for tailoring AMI treatment to individual patients. Cuproptosis, a copper-dependent cell death process characterized by the aggregation of lipid acylated mitochondrial enzymes, is closely associated with disease progression (*Chang et al., 2021*). Cuproptosis and its mechanisms in relation to various diseases have not been studied in great detail. Our goal was to determine the specific role of cuproptosis-related genes in the microenvironment and phenotypes of AMI.

Neither normal subjects nor patients with AMI have previously been studied in order to determine the expression profile of cuproptosis regulators. An elevated CRG level is found more often in AMI patients than in normal individuals, indicating that CRG plays a crucial role in the occurrence of AMI. We calculated correlations between CRG and AMI in order to determine their relationship. In patients with AMI, many cuproptosis modulators exhibit synergistic or antagonistic effects. A mass of immune cells was altered between normal subjects and AMI patients. Patients with AMI showed higher infiltration levels of T cells CD4 memory activated, NK cells resting, macrophages M0, and neutrophils. Additionally, we used unsupervised cluster analysis to identify two distinct clusters associated with different cuproptosis regulators in AMI patients based on CRG-expressed landscapes, and identified two distinct clusters associated with cuproptosis. Cluster 2 exhibited a high immune score and a relatively high level of immune infiltration. Cluster-specific DEGs showed that Cluster 1 was mainly enriched in the ECM receptor and epithelial cells pathways. Cluster 2 had characteristics of mismatch repair, DNA replication, cell cycle, pyrimidine metabolism, RNA polymerase, nucleotide excision repair, and purine metabolism.

It has been reported that endothelial injury is the first line of defense in cardiovascular accidents, and Cluster 1 showed stronger activity on receptors and endothelial cell signaling pathways on the endothelial surface (*Kevil, Patel & Bullard, 2001*). In summary, it is reasonable to believe that Cluster 1 can better assist in the diagnosis of AMI and has a better prognosis. Since cuproptosis can affect metabolic processes such as the TCA cycle and the proliferation, differentiation, survival, and apoptosis of immune cells (*Solomons, 1985*), communication and regulation mechanisms between cuproptosis-related cells and immune cells may not only be helpful in further understanding the pathogenesis of AMI, but may also be a new direction for research.

Predicting the prevalence of AMI using machine learning models based on demographics and imaging indicators has become increasingly popular over the past few

years. According to these studies, multivariate analyses are more reliable and have lower error rates than univariate analyses because they account for the relationship between variables. Our current study compares the prediction performance of four machine learning classifiers (RF, SVM, GLM, and XGB) based on their cluster-specific DEG expression profiles, and established a GLM based prediction model that showed the highest predictive effect in the test cohort (AUC = 0.87). In terms of predicting subtypes of AMI, GLM-based machine learning showed satisfactory performance. We selected two important variables (ZNF302 and CBLB) to construct a GLM based on two genes. ZNF302 is a nickel hydrolysate with sequence specificity based on the Naa-(Ser/Thr)-XAA-HIS-ZAA pattern, with cleavage of the Naa-(Ser/Thr) peptide bond regardless of the Naa properties (considered to be any amino acid or N-terminal peptide sequence) (*Kurowska et al., 2011*). The structure domain of ZNF302 is tetrapeptide (*Kurowska et al., 2011*). ZNF302 is a member of classical Cys2His2 zinc fingers (ZnFs), but its function has yet to be determined. A GWAS analysis of the CHD population revealed that ZNF302 was a significant differential gene (*Jiang et al., 2018*). Therefore, ZNF302 plays a key role in the progression of cardiac development, suggesting that ZNF302 may be a potential diagnostic factor for cardiovascular disease. Casitas B-lineage lymphoma proto-oncogene B (CBLB) is mainly expressed in T cells and macrophages in human and mouse atherosclerotic plaques (*Jeon et al., 2004*; *Romo-Tena et al., 2018*). In bone marrow-derived macrophages, CBLB deficiency increases CD36-mediated lipid uptake, which promotes foam cell formation and the production of inflammatory mediators induced by LPS (*Seijkens et al., 2019*). Additionally, CBLB deficiency induces atherogenic mononuclear and macrophage phenotypes. In atherosclerosis, adaptive immune cells, especially T cells, are recruited (*Vellasamy et al., 2022*). Studies have shown that CBLB can affect the activity of toxic T cells in atherosclerosis (*Li et al., 2018*; *Liu et al., 2015*).

The two-gene-based GLM can accurately predict AMI in two external validation datasets (GSE123342, AUC = 0.856; GSE66360, AUC = 0.719), which provides new insights into the diagnosis of AMI. More importantly, we conducted additional analysis on mice with varying durations of infarction and observed that there was no statistically significant disparity in the expression levels of CBLB and ZNF302 in mice with a 3 h MI. As time progressed, the MI mice exhibited evident MI and impaired cardiac function, accompanied by a significant increase in the expression of CBLB and ZNF302 in the myocardium. Furthermore, it is noteworthy that this indicator exhibits minimal variation in early myocardial tissue. Nonetheless, our study revealed a higher sensitivity of serum results in the MI model, as confirmed through serum mRNA verification. The continued elevation of this index at 48 h and 1 week post-MI suggests its substantial diagnostic utility. These experimental findings provide substantial evidence for the significant upregulation of these two molecules in the myocardial injury model. The progressive increase in these indicators over the course of myocardial infarction from 3 h to 1 week aligns with the observed decline in cardiac function as indicated by cardiac ultrasound. This indicator may serve as a measure of impaired heart function. However, additional research is required to determine whether manipulating the expression of these two genes can mitigate the decline in new functions following myocardial infarction. We further demonstrated that ZNF302

and CBLB play an important role in the diagnosis and treatment of AMI using a mouse model of ischemic myocardial infarction. Therefore, this gene is of great significance for the diagnosis and treatment of AMI.

There are some limitations to this study that need to be highlighted. A comprehensive bioinformatic analysis and mouse experiments have been used in this study, but further clinical testing is required to verify the expression level of CRGs. For the diagnosis model to perform as expected, more detailed clinical features are required. Moreover, we need to explore the relationship between cuproptosis-related clusters and immune response using more AMI samples.

### Funding
This work was supported by the Zhejiang Provincial Natural Science Foundation (No. LQ23H020003) and Ningbo Natural Science Foundation (No. 2022J265). The funders had no role in study design, data collection and analysis, decision to publish, or preparation of the manuscript.

### Grant Disclosures
The following grant information was disclosed by the authors:
Zhejiang Provincial Natural Science Foundation: LQ23H020003.
Ningbo Natural Science Foundation: 2022J265.

### Competing Interests
The authors declare that they have no competing interests.

### Author Contributions
- Bingyu Wang conceived and designed the experiments, performed the experiments, analyzed the data, prepared figures and/or tables, authored or reviewed drafts of the article, and approved the final draft.
- Jianqing Zhou analyzed the data, authored or reviewed drafts of the article, and approved the final draft.
- Ning An analyzed the data, prepared figures and/or tables, authored or reviewed drafts of the article, and approved the final draft.

### Animal Ethics
The following information was supplied relating to ethical approvals (*i.e.*, approving body and any reference numbers):

Committee of Ningbo University School of Medicine (11649) and the use committee and conducted according to the NIH Guide for the Care and Use of Laboratory Animals.

### Data Availability
The datasets supporting the conclusions of this article are available at GEO: GSE48060, GSE66360, GSE123342.

## Supplemental Information

Supplemental information for this article can be found online at http://dx.doi.org/10.7717/peerj.17280#supplemental-information.

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
