# Peer review of "Investigating molecular markers linked to acute myocardial infarction and cuproptosis: bioinformatics analysis and validation in the AMI mice model"

_PeerJ, doi:10.7717/peerj.17280_

## Round 0.1 · original submission · Major Revisions

Thank you for your submission. Three independent reviewers have assessed your manuscript and we are all in agreement that it requires major revisions. If you choose to resubmit, please address all points raised. In particular, all concerns regarding methodology, power, and grammar must be fully addressed.

Reviewer 1 ·

Basic reporting

A. English: While the essential information is present, I would strongly recommend revising the language for a more fluid and easily comprehensible structure with sufficient information. Here are the concerns that I have noticed and addressing these concerns would enhance the overall quality of the manuscript.-- In addition to that, please go through the manuscript carefully to check the logic and languages, typos.
Please note, I only listed some which caught my eye, and there are other which needs to be taken care of to make it more logic or just make sure the information is accurate.

Some of the words need to be defined fist, in order to be readable and accessible to a broader audience, like “cuproptosis," "GSE48060 dataset”.
1. LN43-48, rephrase this sentence to make it logic” Apoptosis, pyrodeath, and ferroptosis are all different mechanisms of cell death, but cuproptosis is unique” what is the definition of cuprotosis? cuproptosis is presented as a unique or distinct form of cell death? ---Define first
2. LN 67. Twenty-one healthy and 31 AMI blood samples were selected from the GSE48060 dataset (GPL570-55999 platform) for further analysis-- Not reasonable English
3. LN72-74. Grammarly check.
4. LN114-116, host species for these antibodies? -Human?
5. 118-122. what is LM22 signature matrix?- maybe it is just because I am not aware of the definition, what LM22 signature matrix represents and how it's used in combination with the CIBERSORT algorithm would enhance understanding?
6. LN173. Definition of DCA?
7.LN189-190, when you first time mention the abbreviation, provide the definitions. (MTF1, ATOX, MAP2MK1, etc)
8. Throughout the manuscript, “copper death” has been mentioned many times, As of my knowledge, the term "copper death" is not a widely recognized or established scientific term in the broader scientific literature, please address the wording.
B. Sufficient literature support, article structure, and relevant results

Experimental design

The manuscript provides insights into the diagnosis of Acute Myocardial Infarction (AMI) through the application of a two-gene-based Generalized Linear Model (GLM). Tow genes were reported here , in which ZNF302 and CBLB emerge as key players in the diagnosis and treatment of AMI and validation was conducted in mouse model. The significant upregulation of these genes in response to myocardial injury establishes them as potential biomarkers and therapeutic targets. I have no doubt that this study contributes to the growing body of knowledge surrounding AMI and opens avenues for further research aimed at refining diagnostic and therapeutic strategies for this critical cardiovascular condition.
In regards to the methods and design, I have some concerns which might to be addressed.
1. The sample size of 31 AMI samples used in your clustering analysis with 46 Cuproptosis-Related Genes (CRGs) ----isn’t relatively small, especially for certain types of clustering algorithms that might require larger sample sizes to identify robust patterns, do you have reference to support your validation and accuracy of your analysis?
3. The title “CBLB and ZNF302 are diagnostic molecules of acutemyocardial infarction in a mouse model”, I understand that the rationale of the manuscript, human-model-prediction- Mouse model validation, but the description of the validation in a mouse model is too general, I would recommend revising by providing more data or at least more description of the mouse model data.

Validity of the findings

My major concern for validity of the findings is listed above:
1. is the sample size big enough in humans.
2. mouse model validation is listed in the title, it needs more clarification.

Additional comments

1.. Section 3.3. LN 218-220, The evaluation of CRG expression differences between the clusters (high expression levels of specific genes in each cluster) provides a molecular basis for the classification.-I I am wondering whether feasible to interpret these differences in the context of cuproptosis and the known functions of the involved genes.-- I do believe the rational of the study is reasonable, but the description of how/why the two clusters and the methods to analyze the CRG expression level assessment is not clear enough- Significant effort are required to clarify the classification and gene expression level.
2. LN189-191, The passage mentions expression levels of genes (MTF1, ATOX, MAP2K1, PDE3B, LIPT1, GLS, DBT) but does not specify the units or scale of measurement (e.g., fold change, log2 fold change) used to determine higher or lower expression levels.

Reviewer 2 ·

Basic reporting

Comments:
1. When referring to the dataset for the first time (in the abstract), please describe the dataset.
2. Exploratory datasets are too small. At least to confirm your findings in one or two other recent datasets.
3. Keep the subtitles in a similar format. Make them clear and include key factors.
4. In the method, please provide the datasets used for CIBERSORT.
5. Lack of full name of *2.10 WGCNA*
6. Would you provide conclusions for each section of results instead of purely describing your observations? e.g., What do the alterations on immune cell landscape mean?

Experimental design

See Basic reporting

Validity of the findings

See Basic reporting

Reviewer 3 ·

Basic reporting

1. There are some grammar issues. Please go through the manuscript carefully. For example, "microarray datasets" should be "microarray dataset" because there is only one dataset used here. The small numbers below ten should be spelled out (line 220).
2. The author does not provide the code of the manuscript to generate the result. I suggest the author provide this information, which is important for reproducibility.
3. The text of many figures cannot be seen clearly (e.g. Figure 2E G, Figure 5F, Figure 9A). I suggest that the author increase the resolution of the figures, especially the figures with text.

Experimental design

One of the most important issues in this manuscript is the authors do not follow the standard machine-learning process.
1. The author does not correctly split the dataset. Usually, the dataset should be divided into a training set and a testing set. To select models or parameters, the training set will be further divided into a training set and a validation set. Using the training set to train the model and using the validation set to validate the models and parameters (cross-validation). Please note that the test set should not be accessed until the models and parameters are selected. However, the author only split the dataset into the training set and the validation set (line 164). I cannot find the definition of the test set (line 257) The performance is overestimated and estimated in this way.
2. Second, the sample set is too small (only 31 samples). I suggest the author perform a leave-one-out strategy for cross-validation to avoid overfitting.

Validity of the findings

The result is overestimated. I suggest the author check the machine learning process and update relevant results.

---

## Round 0.2 · accepted · Accept

Thank you for addressing the concerns raised in the prior review.

Reviewer 1 ·

Basic reporting

I noticed a significant improvement compared to the previous version. In my opinion, this version has clear and unambiguous English, and it follows a professional article structure. It is self-contained and includes relevant results that align with the hypotheses.

Experimental design

I have no doubt that this study contributes to the growing body of knowledge surrounding AMI and opens avenues for further research aimed at refining diagnostic and therapeutic strategies for this critical cardiovascular condition.
This concerns I had regarding the sample size seems to prompt the authors to conduct a validation analysis on two external datasets and incorporate a leave-one-out cross-validation approach to validate their analytical method, and I Do believe that this is a significant step.

Validity of the findings

No comments

Reviewer 2 ·

Basic reporting

The author has made modifications regarding my comments, so I'd suggest accepting the manuscript.

Experimental design

The author has made modifications regarding my comments, so I'd suggest accepting the manuscript.

Validity of the findings

The author has made modifications regarding my comments, so I'd suggest accepting the manuscript.

Reviewer 3 ·

Basic reporting

The issues have been improved.

Experimental design

The issues have been improved.

Validity of the findings

The issues have been improved.